# Post Penetrating Keratoplasty Ectasia: Incidence, Risk Factors, Clinical Features, and Treatment Options

**DOI:** 10.3390/jcm11102678

**Published:** 2022-05-10

**Authors:** Antonio Moramarco, Lorenzo Gardini, Danilo Iannetta, Piera Versura, Luigi Fontana

**Affiliations:** IRCCS Azienda Ospedaliero-Universitaria di Bologna, Via Palagi 9, 40139 Bologna, Italy; antonio.moramarco@aosp.bo.it (A.M.); lorenzo.gardini4@studio.unibo.it (L.G.); danilo.iannetta@unibo.it (D.I.); piera.versura@unibo.it (P.V.)

**Keywords:** corneal transplantation, keratoplasty, corneal ectasia, keratoconus surgery, topography, astigmatism correction

## Abstract

BACKGROUND: Corneal transplantation in keratoconus (KC) patients is generally considered to be successful with a high grade of patient satisfaction. Long-term studies suggest a 6% to 11% probability of KC recurrence manifested by keratometric instability and progressive corneal ectasia. METHODS: We propose to review the frequency, risk factors for the development, and the surgical options for the correction of high irregular astigmatism due to late graft ectasia following penetrating keratoplasty (PK). RESULTS: Post-keratoplasty ectasia is characterized by increasing corneal steepening with myopic shift and high irregular astigmatism, developing years or decades after PK, mostly occurring in KC patients. Contact lenses may adequately improve the visual acuity; however, because these patients are often elderly and intolerant to hard contact lenses, ultimately a surgical correction is proposed to the patient. Compressive suture and corneal wedge resection may improve corneal astigmatism, but the outcomes are unpredictable and often temporary. For this reason, a larger PK graft is often proposed for surgical rehabilitation with the consequence of removing more of the recipient’s healthy endothelium and exposing the patient to a renewed immunogenic stimulus and short-term graft failure for endothelial decompensation. More recently, lamellar keratoplasty using various techniques has been proposed as an alternative to PK in order to maximize the visual outcomes and minimize the complications. CONCLUSIONS: Management of advanced corneal ectasia is a significant challenge for corneal surgeons. Many surgical approaches have been developed, so there is a large arsenal of surgical operations to correct post-PK ectasia. Among them, large-diameter anterior lamellar keratoplasty may be a viable, safer, and effective alternative to PK for the correction of post-keratoplasty ectasia.

## 1. Introduction

Corneal transplantation in keratoconus (KC) patients is generally considered successful with a high grade of patient satisfaction [1]. However, long-term studies suggest a 6% to 11% probability of KC recurrence, several years after surgery [2], manifested by keratometry instability and progressive corneal ectasia. An even higher percentage is reported when patients are followed beyond ten years after final suture removal [3].

In KC patients, increasing irregular astigmatism after penetrating keratoplasty (PK) may result from KC recurrence in the donor button, spontaneous evolution of KC in the host rim with consequent graft–host interface thinning and misalignment, or grafting a donor cornea with undiagnosed KC. Whatever the cause, post-keratoplasty ectasia is characterized by increasing corneal steepening with myopic shift and high irregular astigmatism, developing years or decades after PK, mostly in KC patients [4], often resulting in significant visual impairment.

Because these patients are often elderly and intolerant to hard contact lenses, ultimately, a large PK graft is often proposed for surgical rehabilitation with the consequence of removing more of the recipient’s peripheral healthy endothelium and exposing the patient to a renewed immunogenic stimulus and short-term graft failure for endothelial decompensation.

Herein, we propose reviewing the frequency, development risk factors, and surgical options to correct high irregular astigmatism due to late corneal ectasia after PK.

## 2. Clinical Features and Visual Disturbances

Corneal ectasia is a disorder characterized by the gradual protrusion of the cornea with thinning and steepening, causing progressive visual loss [4]. Causes of corneal ectasia include keratoconus, pellucid marginal degeneration, keratoglobus, keratorefractive surgery, and incisional corneal surgery, i.e., PK. The correct apposition between the graft and the host is an essential goal in PK, representing a prognostic factor affecting the surgical outcome [5]. Paradoxically, a clear corneal graft may be an optical failure if high astigmatism limits the visual result [6].

Corneal ectasia after PK is characterized by decreased uncorrected visual acuity, increased corneal aberrations, and often loss of best-corrected visual acuity. Astigmatism is the most common complication following corneal transplantation, mainly when PK is performed for diseases associated with high preoperative astigmatism, among them, KC [7]. Corneal ectasia after surgery is associated with progressive corneal thinning and irregular myopic astigmatism, so some authors have defined these changes as secondary KC [8]. After PK, despite a clear corneal graft, the eyesight may decrease because of the development of a high degree of astigmatism (more than five diopters (D)), which may occur in as many as 19–38% of surgeries and more frequently in KC patients [9]. In these patients, PK replaces the affected cornea at the time of surgery, but several years later, progressive irregular astigmatism, often requiring surgical correction, may be the result of KC recurrence in the donor button [10], spontaneous evolution of KC in the peripheral host rim with consequent graft–host interface thinning and misalignment [11], or grafting a donor cornea with undiagnosed KC [12]. Whatever the cause, a subtle process of progressive corneal ectasia develops decades after PK producing changes in the refraction and keratometry over time. As a result, the patient’s visual function can be compromised by refractive anisometropia and high postoperative astigmatism. Anisometropia may cause headache, photophobia, burning, tearing, diplopia, and blurred vision. Patients with severe corneal ectasia suffer from varying degrees of ocular aberration, including glare, halos, multiple images, ghosting, reduced visual acuity, and intolerance to prescribed glasses with loss of productivity, lowered self-esteem, and difficulties when performing high-skill visual tasks [4]. On examination, post-PK corneal ectasia resembles many of the clinical features seen in early and advanced stage KC varying from scissors reflex on retinoscopy, commonly seen in early-moderate ectasia, to outward bowing of the lower lid on downgaze (Munson sign), which is a nonspecific finding seen in severe ectasia (Figure 1) [4].

Ectasia is usually preceded by thinning of the recipient’s corneal stroma at the graft–host junction, more frequently located inferiorly and rarely involving the superior quadrant [13]. This is a crucial feature because host thinning adjacent to the graft is often associated with a significant increase in corneal and refractive astigmatism, as demonstrated by the high mean keratometric cylinder [7]. The hallmark of post-PK corneal ectasia is the presence of focal or generalized thinning with protrusion of the graft–host junction [14]. The central corneal graft is usually clear with normal thickness, but in some cases, fine stroma Vogt’s striae, Fleischer ring, and central corneal thinning have been described suggesting the possibility of transmission of undiagnosed keratoconus from the donor cornea [15].

Measurement of intraocular pressure by applanation tonometry (Goldmann, Perkins, etc.) is artifactually reduced due to tissue thinning in ectatic disease. The red reflex should be assessed by examining the fundus to look for a dark area caused by total internal refraction (oil droplet) [4,16].

### 2.1. Topography

The diagnosis of post-PK corneal ectasia is usually based not only on patient history and symptoms, but also on characteristic findings on photo-keratometry, topography, and tomography [17]. Although a unique keratometry value does not define ectasia, photo-keratometry can provide clues in detecting early peripheral abnormalities in the wound interface, showing widening of the mires and greater distance between the rings [13]. Topography allows quantitative measurements of the corneal astigmatism across the whole corneal surface. Repeated exams carry a relevant value in the diagnosis of progression of ectasia with a wide range of astigmatism ranging between 2.5 and 12 D [10]. Astigmatism is usually irregular with asymmetric bowing or with a distinctive pattern. Topography of the anterior surface of the cornea is essential because it forms the foundation of all diagnoses of ectasia based on indices and scores. However, topography does not provide corneal thickness measurements that contribute to diagnosing and managing advanced corneal ectasia (Figure 2 and Figure 3) [8].

Handheld ultrasonic pachymetry remains the standard method for corneal thickness measurements, but the results cannot be related to topography values. There are various tools such as slit-scanning and Scheimpflug imaging that can be used for topographical and combined pachymetry assessment of the cornea. Slit-scanning provides pachymetry values, but repeatability and posterior curvature mapping are inaccurate. Scheimpflug imaging is the preferred modality by most corneal surgeons because it provides precise corneal thickness and posterior elevation with good repeatability [18].

### 2.2. Anterior Segment OCT

Anterior segment optical coherence tomography (AS-OCT) is a valuable instrument that can offer high-definition images of the cornea, angle, anterior chamber, and lens [8]. The newer generation AS-OCT allows real-time high-resolution cross-sectional images topographic and pachymetry mapping across any meridian of choice in ectatic corneal disorders (Figure 4).

Graft–host misalignment and malposition are associated with postoperative astigmatism in most patients. AS-OCT is useful for evaluating the graft–host junction, which is needful for diagnosing and managing complications, such as post-PK astigmatism, occurring at any time point postoperatively. Characteristically, in post-PK ectasia, corneal thinning is particularly pronounced at 1 mm on either side of the graft–host junction, corresponding to the area of maximal corneal steepening [14].

AS-OCT is a fundamental exam for planning the correction of post-PK ectasia. It allows the surgeon to locate and measure the diameter of the donor graft, the extension of the graft–host thinning, and the exact position of the proximal healthy corneal tissue.

## 3. Frequency of Corneal Ectasia after Keratoplasty

After PK for KC, postoperative corneal ectasia has been estimated to occur in 6% to 11% of the patients at 20 to 25 years after surgery [3]. The latency period is, on average, 19 years after PK, paralleling the natural evolution of keratoconus in adolescence [7,18]. According to these data, ectasia following PK is a rare condition that has been described as developing as early as seven years and usually more than ten years following suture removal and increasing up to 20–25-years after surgery. Accordingly, corneal astigmatism continues to worsen during the same time [18]. The corneal graft–host junction typically heals by one year after PK, and the stability of the corneal surface and refraction is achieved from 3 to 4 months after complete suture removal. However, this time period may significantly vary as wound healing is influenced by factors such as patient’s age, systemic disorders (diabetes mellitus and collagen vascular disorders), and topical and/or systemic immunosuppressive medications [19]. For these reasons, any surgical intervention for post-PK astigmatism should be postponed at least 3 to 4 months after complete suture removal or longer in the presence of the above-mentioned clinical factors [19].

## 4. Mechanisms of Delayed Corneal Ectasia after Keratoplasty

PK for KC generally has an excellent long-term prognosis, but some patients may develop recurrence of ectasia years after surgery as PK is usually performed at a young age, and has a high graft survival rate, allowing for a greater chance of KC progression or recurrence [20,21]. Relapse of ectasia after PK has no clear diagnostic criteria, there is still controversy as to whether the corneal ectasia after keratoplasty for keratoconus is a recurrence of keratoconus or a primary ectasia of the donor cornea. The definition of recurrence in previous reports is variable. Niziols et al. defined recurrent KC as increasing astigmatism and thinning within the donor cornea [22]; Pramanik et al. described paracentral corneal thinning and Vogt striae as seen in primary KC [2]. In addition, Patel et al. added to the definition visible anterior bulge and irregular astigmatism [7]. Many studies have been conducted to discover the exact pathogenetic cause, but despite extensive research, the mechanism of progression of this disease remains uncertain [6]. Regarding timing, the recurrence of ectasia in KC patients has been reported to occur typically two decades after PK [14,23]. Indeed, this interval is similar to the time needed for primary KC to evolve, suggesting a continuing evolution of the ectatic disease. As said, the etiology of “recurrent keratoconus” is unknown. However, various theories have been proposed: inadequate excision of keratoconic host tissue during PK, especially for grafts of 7 mm recipient diameter [13]; grafting donor corneas that have subclinical keratoconus; progression of the ectatic disease [24]; release of degradative enzymes from an abnormal host epithelium; occurrence of alterations in Bowman membrane after epithelium–stroma interaction; and infiltration of abnormal host keratocytes into donor tissue with abnormal collagen production [2,6,25,26]. Failure to completely excise the diseased tissue may lead to the progression of the keratoconus in the host with possible involvement of the donor tissue [27]. This appears as progressive thinning of the host and donor tissues, usually occurring inferiorly, with secondary astigmatism that leads to decreased visual acuity and contact lens intolerance in advanced cases. The possibility of grafting a donor cornea with undetected KC is less probable because the chances that patients received donor corneas with subclinical KC are exceedingly small since eye banks anamnestically screen for keratoconus, making it highly likely that recurrence is due to host factors [7]. So, among these hypotheses, KC disease progression seems to be the most plausible explanation: several studies have shown, histopathologically, that abnormal host keratocytes and epithelial cells can migrate, invade, and repopulate the donor graft and cause the pathologic changes seen in keratoconus [13,28]. The continued progression of the pathological process of keratoconus in the host cornea, leading to peripheral corneal thinning, highlights two essential observations [16]. Firstly, it allows the selection of appropriate treatment to reduce astigmatism, i.e., compression sutures at the graft–host interface in peripheral thinning, rather than using incisional or ablative procedures in an ectatic cornea to flatten the steep meridian. In other words, thinning of the host cornea is the cause of the increased astigmatism, and so a “strengthening” rather than a subtractive procedure in the weakened area may represent a logical choice. Secondly, it shows one of the limitations of PK as a treatment for KC where the entire cornea has a tendency to ectasia, and the pathological process continues in the remaining host cornea leading to late onset of astigmatism. The progression of host KC is characterized by inferior corneal elevation, thinning, and ectasia, which can be confirmed by topography [3,14,27]. A continued natural progression of KC in the host rim can cause thinning at the graft–host interface leading to wound slippage and progressive keratometric steepening and astigmatism. De Toledo et al. and Lim et al. found that thinning of the host cornea can occur 14 and 20 years after PK and that thinning can precede signs of recurrent ectasia [11,16]. Another theory is that environment can induce the recurrence of KC [6]. This is possible, because people who have developed KC in the past may still live in environments that originally were partly responsible for the development of KC, such as with high levels of UV light, allergens, or a continued propensity for eye rubbing. Eye rubbing can cause a thinning and loss of rigidity, which may predispose a biomechanically weakened corneal region to cone formation in response to intraocular pressure-distending forces [18].

## 5. Histological Features of Post-Keratoplasty Ectasia

After PK, host keratocytes replace donor keratocytes. As a result of this process, keratocyte density in the donor cornea is lower than in a normal cornea [7]. There is evidence that over the first two decades after PK for KC, donor keratocytes are replaced by a low density of abnormal host keratocytes. Therefore, both the host and donor stroma become thinner at the weak graft–host junction, leading to recurrent ectasia. This hypothesis is supported by the observation that host thinning occurs approximately 14 years after PK [7]. Histopathology exams have revealed marked host cornea thinning and Bowman’s layer disruption or absence of it in both the host cornea and the graft [21], which are also typical findings of primary KC [1,21]. Pathologic studies of failed grafts have shown focally symmetrical central corneal thinning with paracentral breaks in the Bowman membrane, stromal edema, and an intact Descemet’s membrane with a moderate hypocellular endothelium [15]. The histologic and refined analyses of the corneal buttons structure support the hypothesis of recurrent keratoconus because they show a mechanism of slow but continuous development of typical KC characteristics within the donor corneal graft after PK [10]. Moreover, morphologic abnormalities of the Bowman’s layer and characteristic peri-keratocytic arrangement of the stromal deposits are compatible with keratoconus processes [7,14].

## 6. Management

### 6.1. Contact Lens Fitting

The degree of post-PK corneal ectasia may range from mild to severe. Mild cases can be managed successfully with spectacles [29]; nevertheless, irregular astigmatism limits the efficacy of spectacle correction for most patients who often do not tolerate eyeglasses of more than 3 to 4D astigmatism or anisometropia [30]. In these cases, contact lenses (CL) may provide effective correction in 80% of the patients, but CL wearing may be limited by intolerance in the post-keratoplasty population due to ocular surface problems such as dry eye, corneal neovascularization, and corneal morphology alterations [29,31]. Post-PK ectasia can cause severe astigmatism with a highly irregular corneal profile that may hinder CL fitting [23]. There are three different types of CL fitting: apical clearance, apical bearing, or three-point touch [8]. The apical clearance fitting has no contact in the apical area as the lens rests on the periphery. The advantage is that there is a reduced risk of CL-induced corneal scarring, whorl keratopathy, and erosions; the limitation is that tightening at the periphery can impair the tear exchange and that the edge of the lens can be interposed in the visual axis, especially in cases with advanced ectasia. In apical bearing fitting, the optic zone of the CL rests on the apex of the graft. The advantage is that there is a better quality of vision, but the problem is that it may induce corneal scarring and intolerance over the long term. In three-point touch, CL bearing is shared between the apex and the mid-peripheral cornea, which minimizes the risk of corneal scarring. These lenses are the most preferred type because, thanks to their features, they can provide good vision, better comfort, and prolonged wearing time. Corneal surgeons prefer to prescribe CL in cases with regular astigmatism of more than 3.0 D, irregular astigmatism, anisometropia, or aniseikonia [32]. However, corneal abnormalities, dry-eye syndrome, fitting-related difficulty, and lifestyle problems are significant concerns that may affect a patient’s CL tolerance. Other essential parameters that must be considered are the diameter of the graft, the topographical relationship between the host cornea and donor cornea, the graft toxicity, and the location of the graft.

Post-keratoplasty astigmatism can be managed with various CL types: hybrid, scleral, rigid gas-permeable, or piggy-back CL.

Rigid gas permeable lenses are considered successful in managing irregular astigmatism encountered with corneal ectasia. They have an anterior and a posterior surface, creating a spherical anterior refractive surface over the ectatic cornea. In addition, tear film provides an optical bridge between the posterior aspect of the CL and the irregular cornea. Up to 80% of ectatic eyes may be successfully managed with rigid gas permeable lenses alone [33]. They are considered to be the gold standard of irregular cornea visual rehabilitation, but rigid gas permeable CL may not always be appropriate because variable curvatures of the ectatic area both centrally and at the donor–host junction may be the reason for lens decentration, instability, or even dislocation with blinking. Other disadvantages are that a rigid CL glides over an irregular or highly toroid corneal surface, leading to corneal micro-trauma, epithelial and anterior stroma disruption, and chronic ocular inflammation. Furthermore, the thickness of these lenses reduces gas transmission, so insufficient oxygen supply may cause corneal hypoxia with an increased risk of graft edema.

Another option is scleral CL, which is known to improve visual function in patients with corneal irregularity [33]. Their primary characteristic is that they do not touch the cornea and limbus as they rest on the sclera, leaving a clear area between the CL and the cornea. The advantages provided are good centration, stability, and improved visual acuity [8]. The development of rigid gas-permeable plastics has significantly reduced the hypoxic complications associated with daily wear of CL. In this way, scleral lenses can improve vision in patients with markedly irregular corneas.

In addition to existing full-size scleral lenses [34], other designs, such as corneoscleral (12.5–15.0 mm in diameter) and mini-scleral (15.0–18.0 mm), have been developed. Typically, mini scleral lenses exhibit minimal corneal clearance with small areas of scleral bearing. Taking a break during the day is very important for the performance of all scleral lenses. Indeed, graft patients should be advised to have pauses during CL wear to minimize toxicity damage.

Hybrid CL have been developed to improve fitting tolerance in patients who cannot tolerate rigid gas permeable lenses. Hybrid lenses (i.e., SynergEyes, SynergEyes Inc., Carlsbad, CA, USA) consist of a rigid center with a soft skirt. This combines the advantage of good visual acuity provided by rigid lenses with the comfort of a soft CL. Other types are tandem CL composed of a soft silicone hydrogel lens fitted between the rigid gas permeable lens and the cornea, improving comfort and fitting. Tandem lenses can be used initially as lens adaptation, or they may be used for long-term management [29].

### 6.2. Corneal Sutures

In early cases of post-PK ectasia, sutures may help remodel the corneal curvature by placing multiple interrupted tight sutures at the ectatic graft–host junction in correspondence to the maximum steepening meridians. Sutures compress the ectatic graft–host tissue, inducing peripheral flattening and steepening of the central corneal curvature in the axis where they are placed. In order to maximize this effect, limited opening of the graft–host junction in correspondence to where sutures are positioned is recommended. The use of compression sutures alone to control postoperative astigmatism after PK has been reported by Roper-Hall and Atkins [35]. Sutures must be placed at a depth of approximately 50% of the thickness of the cornea. The most significant regression occurs over the first week after suture placement. Applying compression sutures to treat astigmatism after keratoplasty is safe and effective and has a minimal risk of penetration into the anterior chamber. Compression sutures for post-PK ectasia have been described alone [36] or in combination with arcuate relaxing corneal incisions [37]. The combined use of these techniques may produce a more significant diopter shift, but the risk of perforation and the unpredictability of the astigmatic change must be considered [38].

The advantage of this correction is that it is safe and reversible, but the limitation is that improvement may be unpredictable and subject to loss of effect with time due to the tissue elasticity and progression of the ectatic development.

### 6.3. Wedge Resection

If ectasia is confined to a limited extension along the graft–host junction, a wedge resection can be performed in the same area to reduce astigmatism [23]. Wedge resection aims to remove the abnormally distended host tissue, correcting the corneal shape and astigmatism in post-PK ectasia [39]. This technique requires an initial overcorrection of more than 50% and inelastic (nylon or prolene) compressive sutures to minimize regression of the effect with time (Figure 5).

Furthermore, removing the diseased tissue may prevent or delay the need for repeat PK. For this reason, this procedure is often considered before planning a repeat PKP for high astigmatism secondary to “recurrent” KC [27]. A wedge of corneal tissue is excised from the donor and host cornea along the flattest meridian, manually, with a metal or diamond knife or using a femtosecond laser. Following excision, the “shortened” tissue is apposed with tight sutures inducing corneal steepening. Several factors, including the excised corneal tissue’s width, length, depth, and location, can affect the refractive outcome [32]. In particular, the second incision is more difficult because, after the first incision, the cornea is more flaccid. Another problem with this technique is excising the exact amount of tissue in depth and width. Recently, a femtosecond laser has been introduced to facilitate corneal wedge resections [32]. The laser-assisted procedure is more precise and accurate, increasing predictability compared with the manual technique. Wedge resection is reserved for grafted eyes with high degrees of astigmatism, usually over ten diopters. This technique has several advantages over a repeat PK, such as preservation of the clear central cornea, no risk of rejection or interface haze, better wound strength, and a shorter visual rehabilitation period [8]. Disadvantages are postoperative unstable astigmatism because of the persistent tension at the sutured wound and recovery over six months; in addition to concerns regarding the stability of the surgical results in the long term.

### 6.4. Intra Ocular Lens Implantation

Toric Intra Ocular Lens (IOL) has been described to correct post-PK astigmatism in phakic eyes during cataract surgery. IOLs can provide a wide range of options for correcting spherocylindrical errors and do not require significant manipulation of the grafted tissue [32]. A potential limitation is a surgically induced astigmatism (SIA) resulting from toric IOL insertion, especially with the rigid polymethyl methacrylate IOL implantation through a 6 mm incision. Studies reported a mean SIA of 1.85–2.53 D after toric IOL implantation following keratoplasty [30,32]. With the development of new foldable toric IOLs, which require smaller incisions, there will be less SIA. Concerns about this treatment modality are endothelial cell loss and SIA, which are less predictable in grafted corneas. Still, it can be acceptable as this technique aims to reduce post-keratoplasty astigmatism [19,32].

The Artisan toric (Artisan, Ophtec BV, Groningen, The Netherlands) AC IOLs provide a good range of solutions for the correction of post-keratoplasty astigmatism and ametropia for both phakic and aphakic eyes. Major concerns with this procedure are potential complications such as SIA due to the large implantation incision, endothelial cell loss due to the AC implant, chronic inflammation due to chronic damage of the aqueous blood barrier, and cystoid macular edema. The toric lens is iris-claw fixated and has a 5 mm optical zone that requires a 6 mm limbal incision for implantation. The stability of the postoperative refractive cylinder after Artisan toric lens implantation for up to 36 months was reported excellent [40]. A potential limitation of the Artisan toric IOL is SIA, which may be somewhat unpredictable in post-PK patients with irregular astigmatism [30].

The Intraocular Collamer lens (ICL) (Staar Surgical, Monrovia, CA, USA) is a posterior chamber phakic intraocular lens (pIOL), and it is reported to be effective for the correction of moderate to high myopia, hyperopia, and astigmatism. Concerns have been raised, because after ICL implantation, mechanical contact and rubbing between the ICL and the posterior iris surface can increase pigment dispersion, leading to pigmentary glaucoma and the induction of synechiae iris sphincter erosion, and iris transillumination. Laser iridotomy or surgical iridectomy and adequate ICL sizing are essential to prevent angle-closure glaucoma and pigmentary glaucoma. In addition, ICL implantation may provide an effective alternative for correcting anisometropia and astigmatism after corneal transplantation [31,41].

Another approach for correcting residual pseudophakic astigmatism induced by post-PK ectasia is the piggy-back IOL technique, which minimizes the risk of complications such as zonular, capsular, and corneal endothelial damage and provides more predictable results than in cases undergoing cataract surgery or clear lens extraction for high astigmatic correction. With this technique, a posterior chamber in the bag IOL replaces the cataractous or clear lens to correct the spherical refractive error, and a purposely designed piggyback IOL is implanted in the sulcus to correct the astigmatic error. Sulcoflex (Rayner Intraocular Lenses Ltd., Worthing, UK) and off-label use of ICL have demonstrated good refractive outcomes to correct pseudophakic astigmatism as a piggyback lens. The major advantage of this technique is the reversibility of the implant and the possibility of rotating the piggyback IOL after time to adjust the astigmatic correction. Complications to be considered are endothelial cell loss, ciliolenticular block, and pigment dispersion [42].

### 6.5. Repeat Penetrating Keratoplasty

A large diameter (8.5–9.5 mm) PK is often the treatment of choice in patients with extensive ectasia involving the graft–host junction. Still, numerous disadvantages are associated with this procedure, such as an increased risk of graft rejection, late endothelial failure, cataract development, and augmented risk of postoperative glaucoma [23]. In addition, whenever a new PK is performed, the previous graft is replaced by an unknown donor, which may likely stimulate immunologic rejection. For these reasons, long-term postoperative immunosuppressive treatment is advised in these patients. Historically, repeat PK has a poorer survival prognosis than first-time keratoplasties [43]. Weisbrod et al. [44] reported a significant difference in the five-year survival between first and repeat keratoplasties, 64.6% and 45.6%, respectively. Repeat PK share, on the one hand, the same risk factors for graft failure as the initial keratoplasty concerning the recipient diagnosis. On the other hand, additional risk factors have been acquired since the original procedure, such as increased age in all cases, glaucoma escalation, development of neo-vascularization, and worsening of ocular surface disease in many cases. Moreover, there is a chance of recurrent ectasia even after a second PK, and repeating a corneal transplant in a patient with high and or irregular astigmatism may not necessarily guarantee better results regarding astigmatism [45]. Eccentric placement of the graft can produce astigmatism with the flatter axis in the direction of the displacement. After suture removal, the curvature may change in an unpredictable and often unfavorable manner, even after primary PK. Additionally, with repeat PK, host rim instability may be exacerbated by incomplete excision of the previous host–graft junction in severely decentered grafts. So, with all sutures in place, BCVA and astigmatism improve significantly after large PK as sutures are placed more peripherally, and influence less the central graft. However, to prevent significant recurrence of astigmatism, final suture removal should be postponed as long as possible. Thus, preoperative counseling of keratoconus patients must stress that repeating a corneal transplant in an eye with high irregular astigmatism will not necessarily yield a better astigmatic long-term result, though improved vision might be expected after surgery.

### 6.6. Tuck in Lamellar Keratoplasty

Post-PK astigmatism with thinning of the cornea that involves the periphery makes surgical management exceptionally challenging. Tuck-in lamellar keratoplasty (TILK) is a surgical procedure that can be used for the management of cases with advanced corneal ectasia involving the corneal periphery and the graft–host interface [46,47]. Using a vacuum trephine, a partial-thickness groove of 200–250 mm, depending on the corneal thickness, is made at least 1 mm beyond the graft–host junction to encompass the area of corneal thinning and ectasia. Then, a centripetal stromal dissection is carried out using lamellar dissectors to excise a central anterior stromal disc, paying attention not to cause micro-perforations in the thin residual corneal stromal bed. Subsequently, a centrifugal stromal lamellar dissection is performed using a crescent knife to create a peripheral intrastromal pocket up to 0.5 mm posterior to the limbus. Then, a large 9–9.5 mm diameter lamellar donor graft with a 2 mm tapered peripheral flange is implanted onto the residual stromal bed. Finally, the graft flange is tucked into the previously mentioned peripheral intrastromal pocket, and interrupted sutures are applied to secure the graft onto the host bed. In this technique, the donor lenticular has a full-thickness central part and a peripheral flange consisting of the partial-thickness posterior corneal stroma. Thus, the central part of the full-thickness graft provides tectonic support to the central cornea. With this technique, the thin peripheral flange tucked into the intrastromal pocket provides tectonic support to the weakened peripheral cornea beyond the previous graft–host junction. The dissection of the superficial limbal region is avoided so that there is no damage to the recipient’s limbal stem cells. This technique provides tectonic support and optimal thickness to the peripheral corneal tissue without replacing the original PK graft. A subsequent central optical PK may be performed to achieve adequate visual acuity should it be required.

### 6.7. Peripheral Reconstructive and Annular Lamellar Keratoplasty

Malbran et al. [28] proposed a peripheral reconstructive annular lamellar keratoplasty (PALK) as a customized reinforcement grafting technique to preserve the previous PK and restore normal peripheral corneal thickness. This technique aims to reinforce the ectatic peripheral graft–host junction in the presence of a clear and functioning PK. An 8 to 11 mm lamellar resection is made starting at the outer margin of the previous PK. A same-size full-thickness donor graft (annular or segmental) is sutured, covering the ectatic junction and thin adjacent host rim. A thin stromal lamellar customized resection that starts at the donor/recipient junction and proceeds to the periphery in an annular or crescent shape according to the ectasia configuration is performed. A full-thickness (10–11.5 mm) annular or crescent graft, tailored according to the shape and width of the previously resected area, is sutured independently at the peripheral cornea and the donor/recipient junction; thus, the previous PK clear graft is always fully respected. PALK is a surgical treatment that improves ectasia both functionally and structurally. Another approach was described by Cheng et al. [48], who proposed a surgical technique named C-shape lamellar keratoplasty to manage peripheral degeneration. Once manually outlined with a fine surgical marker, the exact extent of peripheral ectasia, circular trephines of standard diameters are inked to mark the inner and outer circumferential limits of the arcuately shaped ectasia. Commonly, trephines of 9 mm diameter for the internal circumference and 14 mm diameter for the external rim are used. The width between these two marked arcs then is measured using calipers, and disposable dermatological trephines (Kai Dermal Biopsy Punch; Kai Europe GmbH, Solingen, Germany) of corresponding sizes (usually 2 or 3 mm) are used to refine the ends of the C-shaped area of dissection. A vertical incision of the donor and host cornea is performed with a diamond knife, following the C-shaped outline, avoiding accidental perforation. Using a crescent blade, careful lamellar stromal dissection is carried out, ensuring uniform depth of the host bed at the most ectatic areas. Then, a C-shape donor lamellar graft of equal size as the recipient is excised from a donor corneoscleral rim mounted on an artificial anterior chamber. Typically, the width between the two circumferential arcs is deliberately smaller by approximately 0.25 to 0.5 mm than the recipient bed. Finally, the graft is sutured in place with interrupted, tight, compressive sutures, using a combination of 10/0 and 9/0 nylon sutures. The technique of C-shaped lamellar keratoplasty compression aims to minimize forward protrusion of the cornea. The compression is achieved by the deliberate under-sizing of the donor lamella width, resulting in compression and flattening of the peripheral ectasia with and without sutures. Conversely, if the donor grafts were the same size or oversized, less compression and flattening of the ectasia would occur, and significantly more residual astigmatism would be expected. The main problem related to this technique is peripheral vascularization with early loosening of sutures. However, there is no allograft rejection due to the small amount of donor tissue involved. This technique is technically challenging to perform, but it allows for correcting tectonically corneal thinning, reducing the possibility of subsequent perforation, ectasia, and irregular astigmatism, and allows for early visual rehabilitation.

### 6.8. Overlay Deep Anterior Lamellar Keratoplasty (DALK)

Overlay deep anterior lamellar keratoplasty (DALK) for ectasia after PK consists of a large diameter (9–10 mm) lamellar stromal dissection encompassing the PK graft and the thinned graft–host junction, leaving behind the Descemet membrane and a thin layer of the stroma. Then, a same diameter full-thickness donor graft or a stromal lenticule of 300–400-micron thickness, prepared using a microkeratome, is sutured in place. With this procedure, once a cornea of correct profile and thickness has been reestablished, a smaller central PK can be performed should visual acuity results be inadequate due to an irregular graft–host interface [49]. This technique was introduced by Lake et al. [50], who performed manual dissection DALK across the PK graft followed by full-thickness donor graft implantation to provide sufficient stromal tissue for a secondary smaller diameter PK. Performing DALK over PK is technically more challenging than conventional DALK for KC. More attention has to be paid, especially when dissection is performed across the graft–host junction, which is a potential site for macro-perforation. Scorcia et al. [51] applied a modified version of the DALK small bubble technique to treat post-PK ectasia. With this technique, following partial-thickness trephination, a manual stromal dissection extends beyond the PK margins. A small air bubble (5–6 mm) is injected between the pre-DM and stroma in the central PK to facilitate deep stromal excision with the baring of the pre-Descemet’s layer (Figure 6).

This allows the creation of a main smooth recipient bed, favoring the formation of a clean graft–host interface that avoids the potential need for a subsequent central PK. Further modification of this technique was proposed by Yu et al. [52], who described the identification during stromal dissection of a natural plane of separation occurring between the pre-Descemet’s membrane and stroma. This facilitates the excision of the stroma across and beyond the PK graft and allows a regular, clear plane for lamellar graft implantation, minimizing the risk of interface opacity. All the techniques mentioned above share the common advantage of preserving the globe integrity, and preserving donor graft and peripheral host endothelium, thus reducing the risk of endothelial rejection, late graft failure, and complications related to open-sky surgery. Controversy regarding the worth of sparing the “old” PK endothelium that may frequently present a low endothelial cell count has been debated. DALK overlay may correct the donor and host cornea profile and thickness without altering the PK graft transparency while providing a scaffold for a future endothelial keratoplasty should the original PK endothelium fail to maintain a clear graft. As an alternative, Patel et al. [53] evaluated the feasibility of a microkeratome-assisted superficial anterior lamellar keratoplasty (SALK) after PK. This procedure may otherwise be called automated lamellar therapeutic keratoplasty (ALTK). First, an external lamella (free cap) is cut from the recipient cornea with a microkeratome (Moria, Antony, France) using a 130 mm microkeratome head. Then, a stromal lamella of the same thickness is obtained from a donor cornea and secured in place using two overlay 10–0 nylon sutures. The advantage of this technique is that microkeratome-assisted dissection obtains a smooth stromal bed surface compatible with good postoperative visual acuity. Gutfreund et al. [54] proposed a modification of the microkeratome-assisted anterior lamellar keratoplasty, which optimizes the refractive outcome, while minimizing surgical trauma to the recipient endothelium of post-PK eyes with extremely high astigmatism. First, a superficial lamella (free cap) approximately 9.0 mm in diameter is cut from the recipient cornea using a micro-keratome 250 mm head (Moria, Antony, France). Two full-thickness incisions are created through the PK scar using a 15-degree blade to release asymmetric scar tension in the residual recipient bed. Each incision is centered on each side of the steeper meridian. Then, an anterior stromal lamella is cut from a donor cornea using the same 250 mm micro-keratome head used for the recipient. The donor tissue was then punched to the same size as the recipient bed and sutured with 10–0 nylon sutures (Figure 6). The major limitation to the microkeratome-assisted SALK procedures is that they cannot be safely applied to post-PK corneas with severe thinning and ectasia due to the risk of wound dehiscence with vacuum suction during the microkeratome cut.

## 7. Prevention

### 7.1. Crosslink (CXL)

Collagen crosslinking (CXL) exploits riboflavin as a photosensitizer followed by ultraviolet-A light. Collagen fibers in the cornea respond to treatment by developing chemical covalent bonds by photopolymerization, increasing corneal rigidity. Most crosslinking occurs in the 300–350 μm anterior stroma within the central 8–9 mm cornea. According to different trials [55,56], this treatment has been demonstrated to be safe and effective in halting the progression of keratoconus [57]. Some authors have hypothesized that CXL extended to the corneal periphery a few months before keratoplasty may reduce the incidence of recurrent ectasia by strengthening the peripheral corneal tissue [58]. This hypothesis has yet to be confirmed, but CXL performed before and maybe sometime after surgery may represent a potentially useful treatment to prevent the long-term development of ectasia after keratoplasty. Unfortunately, there are not enough data in the literature that support this theory.

### 7.2. Large DALK

DALK is an alternative surgical procedure to PK to treat various corneal pathologies, among them KC, without the involvement of the endothelium. Several studies demonstrated that DALK provides comparable visual outcomes to PK [59] with the advantage of a lower risk of immunologic rejection and long-term graft failure. Large-diameter PK has been shown to reduce postoperative astigmatism and improve visual acuity in corneal ectasia patients, i.e., KC [60]. Similarly, large-diameter (8.5–9 mm) DALK (L-DALK) has proved effective in reducing the degree of postoperative myopia and manifest astigmatism with improved visual outcomes, compared to standard size DALK (S-DALK) in Asian KC patients [60]. L-DALK may provide short-term better visual outcomes and may also play a protective role toward the risk of late postoperative ectasia due to the more extensive pathological tissue excision and to the stronger wound healing induced by the proximity of the graft–host junction to the vascularized corneal limbus. One concern is the risk of potentially increased immunological rejection secondary to conversion to PK due to accidental perforation occurring during the DALK procedure. Indeed, full-thickness grafts larger than 8 mm are associated with an increased risk of endothelial immunologic rejection [61]. In this regard, Myerscough et al. [62] have recently demonstrated that cases of large intraoperative perforation that occurred during L-DALK that were converted to mushroom shape PK showed an excellent visual outcome and low risk of graft failure at five years after surgery. Due to the more recent introduction of DALK in the treatment of KC, long-term data regarding the frequency of post-DALK ectasia are still lacking, and large retrospective studies spanning 15–20 years are required to answer the question if KC patients after DALK may be at less risk of post-keratoplasty ectasia than after PK.

## 8. Conclusions

Management of advanced corneal ectasia is a significant challenge for any corneal surgeon. Over the last decade, many surgical approaches have been developed, so there is a large arsenal of surgical operations to correct post-PK ectasia (Table 1). However, each technique has advantages and disadvantages, so none of them appears as a perfect option (Table 2). Corneal surgeons should tailor a specific plan based on patients’ needs and clinical situations to take advantage of each intervention (Figure 7). So, prevention of progression at an early stage of ectasia appears to be the best approach to deal with these disorders.

Furthermore, there are, so far, only a few anecdotal [63] cases of post-DALK ectasia. For this reason, L-DALK is believed to be the best approach in treating keratoconus at the advanced stage [64]. However, should the clinical situation require a PK solution, a large repeat PK graft, with complete excision of the abnormal graft–host junction, offers a more rapid visual recovery and predictable visual outcome with the known disadvantages of exposing the patient to the risk of serious complications such as immunologic rejection, graft failure, cataract development, and secondary glaucoma due to peripheral anterior synechiae formation. An L-DALK overlay graft may represent a viable and effective alternative to PK for the correction of post-keratoplasty ectasia, offering all the advantages of an anterior lamellar surgery procedure and closed eye surgery.

## Figures and Tables

**Figure 1 jcm-11-02678-f001:**
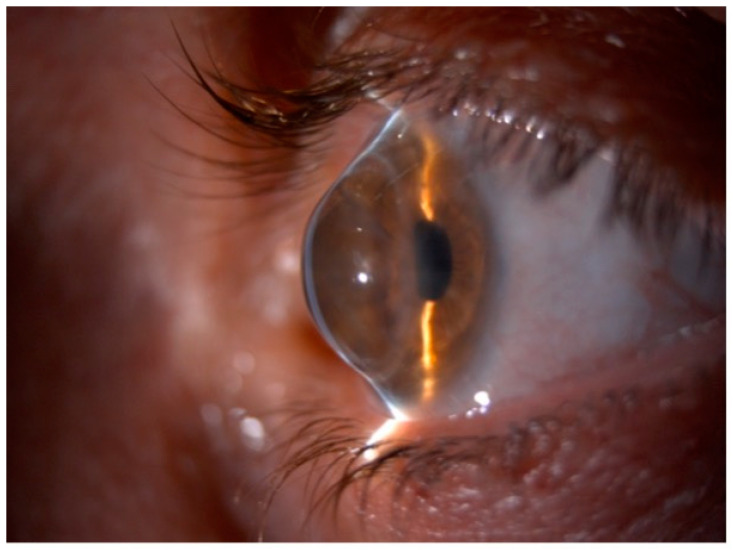
Slit-lamp photograph of post-penetrating keratoplasty ectasia. The patient underwent keratoplasty for keratoconus 23 years before.

**Figure 2 jcm-11-02678-f002:**
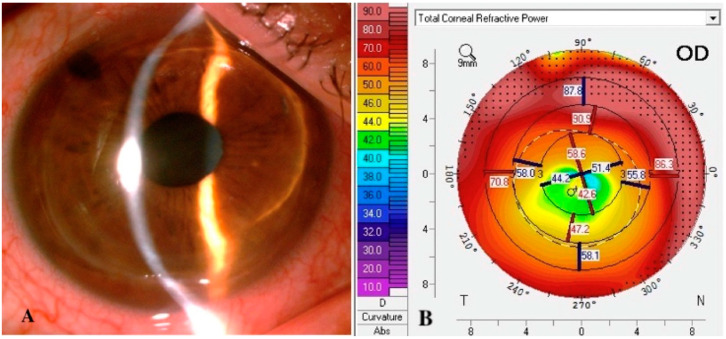
Slit-lamp photograph (**A**) and corneal topography (**B**) of diffuse corneal ectasia after penetrating keratoplasty for keratoconus.

**Figure 3 jcm-11-02678-f003:**
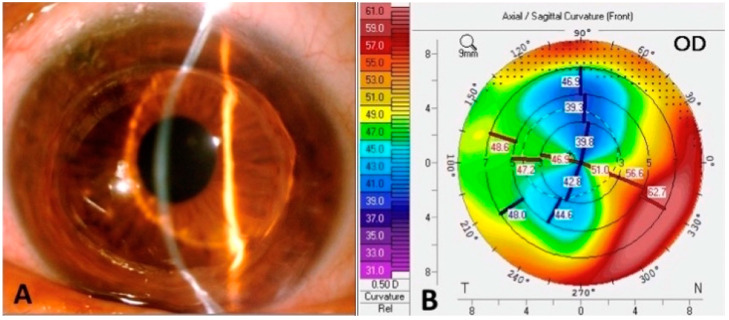
Slit-lamp photograph (**A**) and corneal topography (**B**) of focal inferior corneal ectasia after penetrating keratoplasty for keratoconus.

**Figure 4 jcm-11-02678-f004:**
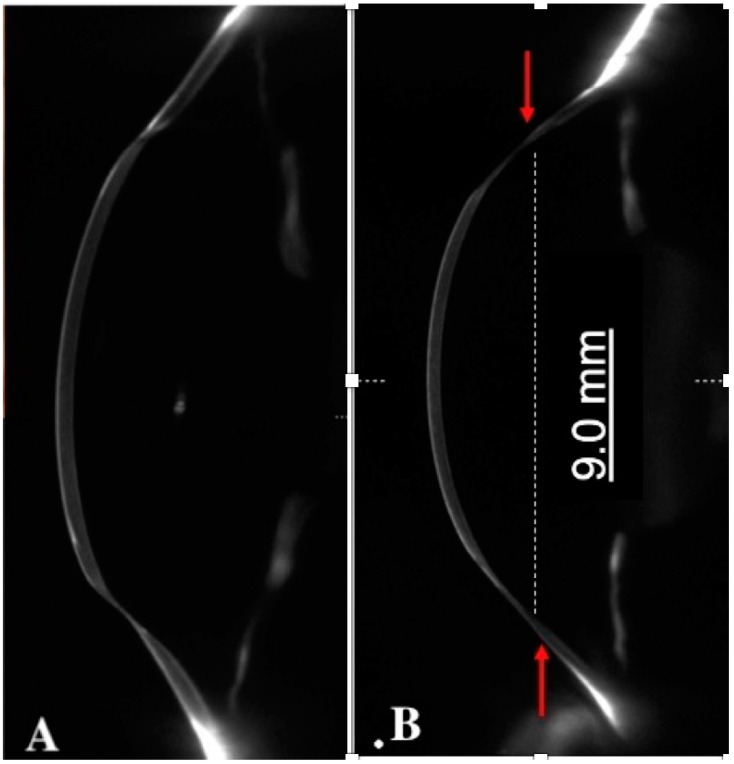
AS-OCT showing focal inferior ectasia (**A**) and diffuse ectasia long time after penetrating keratoplasty. Red arrows indicate peripheral limits of corneal thinning (**B**).

**Figure 5 jcm-11-02678-f005:**
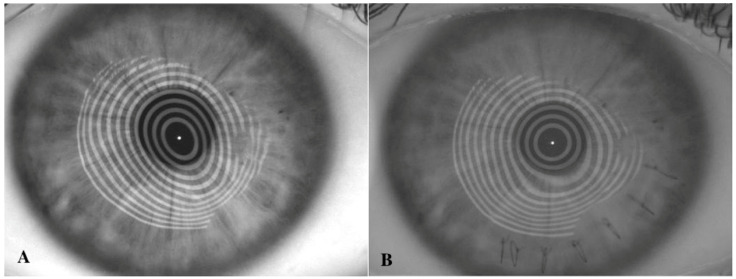
Placido image of pre (**A**) and post-operatory (**B**) of cornea ectasia treated with wedge resection and compressive sutures.

**Figure 6 jcm-11-02678-f006:**
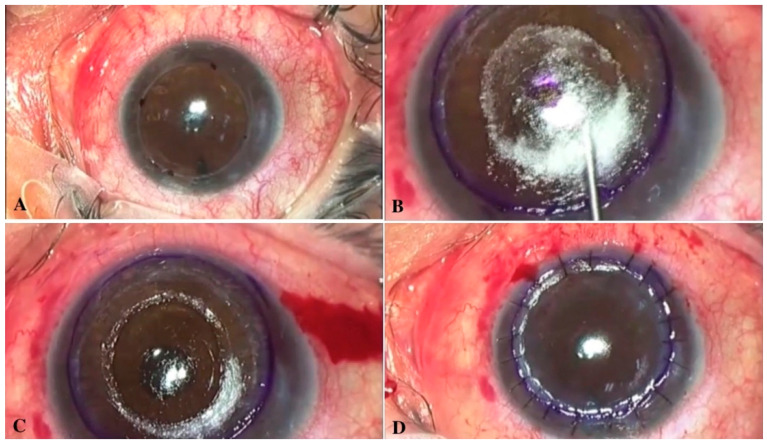
Overlay, deep anterior lamellar keratoplasty technique: (**A**) marking of the trepanation site; (**B**) separation of the stroma using big bubble technique; (**C**) clearing the central zone; (**D**) flap positioning.

**Figure 7 jcm-11-02678-f007:**
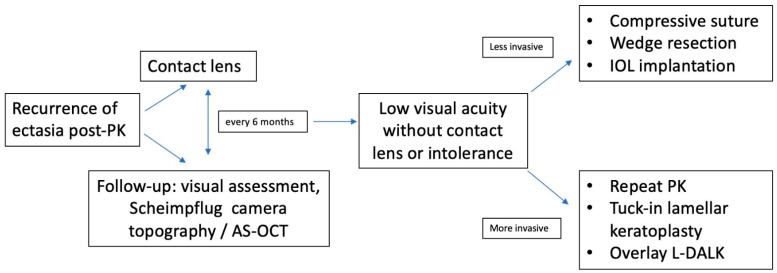
Algorithm for management and correction of post penetrating keratoplasty corneal ectasia. AS-OCT = Anterior segment optical coherence tomography; IOL = intraocular lens; PK = penetrating keratoplasty; L-DALK = Large-diameter deep anterior lamellar keratoplasty.

**Table 1 jcm-11-02678-t001:** Surgical procedures for the correction of post-penetrating keratoplasty ectasia. The number of eyes, the outcomes of the procedures are summarized.

	Procedure	Operated Eyes	Results
Limberg et al.	Compressive suture	5	A 58% reduction of average cylinder in 1 week; reduction of 41% of average cylinder at 9–12 months
Roper-Hall and Atkins et al.	Compressive suture	1	8 compression sutures to treat 10 D of refractive cylinder.
Kirkness et al.	Compressive suture & arcuate keratotomy	38	Reduction of 3.6 D–5 D in astigmatism
Javadi et al.	Compressive suture	77	A mean reduction of 3.4 D and 5.9 D in keratometric astigmatism
Mejia et al.	Wedge resection	39	Astigmatic cylinder improved from 7.99 D to 2.5 D at 12 months and remained stable thereafter
Ilari et al.	Wedge resection	3	Preoperative astigmatism decrease and corneal profile regularized
Nuijts et al.	IOL implantation (Artisan toric lens)	16	Spherical equivalent decreased from 5.50 D before surgery to 0.86 D at final follow-up
Tahzib et al.	IOL implantation (Artisan toric lens)	36	Spherical equivalent decreased from 4.31 D preoperatively to 1.20 D at the last follow-up
Akcay et al.	IOL implantation (Toric ICL)	1	Patient manifest refraction improved from −8.0/−1.75 at 170° preoperatively, to −0.75/−0.50 at 130° postoperatively
Alfonso et al.	IOL implantation (Posterior chamber phakic intraocular lenses)	15	The mean postoperative spherical equivalent was 0.95 D
Al Mezaine et al.	Repeat PK	210	29.6% of the eyes achieved a final visual acuity greater than 20/200, while only 4.8% reached 20/40 or better
Daniel J. Weisbrod et al.	Repeat PK	116	2 and 5-year survival rates for repeat PKP were 63.9% and 45.6%, respectively
Szentmàry et al.	Repeat PK	17	Visual acuity improved significantly from 0.2 to 0.5
Vajpayee et al.	Tuck In Lamellar Keratoplasty	12	Kmean decreased, and the mean spherical equivalent (SEQ) refractive error decreased
Yang et al.	Tuck In Lamellar Keratoplasty with an lenticule	3	The mean min K value measured by topography preoperatively and 1 day, 1 month, 3 months and 12 months postoperative was 44.86 ± 3.92, 48.83 ± 2.38, 50.63 ± 4.24, 49.71 ± 3.26 and 50.07 ± 3.43 respectively. The max K value increased significantly after the operation and then declined gradually
Malbran et al.	Peripheral Reconstructive Lamellar Keratoplasty	33	Preoperatively, the mean keratometric measures were K1: 44.8 D and K2: 54.1 D and postoperatively K1: 47.5 D and K2: 50.8 D.
Cheng et al.	C-shaped lamellar keratoplasty	3	All eyes achieved Snellen visual acuity of 20/40 or better and stable astigmatism ranging from 0 to 2.75 diopter cylinder within 6 months
Lake et al.	DALK Overlay	7	At 12 months, mean UCVA improved from 1.157 to 0.74 Logmar. Mean BCVA improved from 0.82 to 0.37 at 12 months
Scorcia et al.	DALK Overlay (small bubble)	9	BCVA improved by at least 4 Snellen lines
Patel et al.	Microkeratome-Assisted Superficial Anterior Lamellar Keratoplasty	9	BCVA improved in all 9 eyes at final follow-up. Refractive astigmatism also improved by an average of 0.7 diopters.
Gutfreund et al.	Microkeratome-Assisted Anterior Lamellar Keratoplasty (MALK)	4	3 years after MALK, BCVA improved to 20/20, refractive astigmatism decreased of an average of 2.1 D (in all cases within 4.5 D), and the average surface asymmetry index decreased from 2.27 to 0.56

**Table 2 jcm-11-02678-t002:** Types of surgical procedures, indications, advantages and disadvantages, according to the studies cited.

Procedure	Indications	Advantages	Disadvantages
Contact lens fitting	First line treatment	Non-invasive, better visual acuity than spectacles	Corneal scarring, whorl keratopathy, corneal micro-trauma, epithelial and anterior stroma disruption, and chronic ocular inflammation
Compressive sutures	Early case of ectasia post-PK	Minimal risk of penetration into the anterior chamber	Improvement may be unpredictable and subject to loss of effect with time due to the tissue elasticity
Wedge resection	Ectasia of limited extension along the graft–host junction.	Prevent or delay the need for repeat PK, no risk of rejection or interface haze	Postoperative unstable astigmatism.
Intra ocular lens implantation	Correct post-PK astigmatism in phakic eyes or during cataract surgery	Does not alter corneal profile and transparency	Endothelial cell loss, chronic inflammation, cystoid macular edema, pigment dispersion, leading to pigmentary glaucoma, cilio-lenticular block, iris synechiae, sphincter erosion, and iris transillumination
Repeat penetrating keratoplasty	Extensive ectasia involving the graft-host junction	Visual acuity and astigmatism improve significantly after large PK as sutures are placed more peripherally and influence less the central graft	Increased risk of graft rejection, late endothelial failure, cataract development, and augmented risk of postoperative glaucoma and immunologic rejection
Tuck-in lamellar keratoplasty	Diffuse thinning of the peripheral cornea with advanced corneal ectasia involving corneal periphery and the graft-host interface	Tectonic support to the weakened peripheral cornea beyond the previous graft–host junction, no damage to the recipient’s limbal stem cells	Challenging technique to perform for both donor and host preparation
Peripheral reconstructive and annular lamellar keratoplasty	Diffuse thinning of the peripheral cornea with advanced corneal ectasia	Preserve the previous PK and restore normal peripheral corneal thickness, minimize forward protrusion of the cornea	Causes peripheral vascularization with early loosening of sutures. Surgically challenging to perform for both donor and host preparation
Overlay deep anterior lamellar keratoplasty	Extensive corneal ectasia. Aims to correct the donor and host cornea profile and thickness without replacing the PK endothelium	Preserve the globe integrity, the donor graft and peripheral host endothelium, thus reducing the risk of endothelial rejection, late graft failure, and complications related to open-sky surgery	Technically more challenging than conventional DALK, risk of perforation when dissecting across the host-graft junction

PK = penetrating keratoplasty; DALK = deep anterior lamellar keratoplasty.

## Data Availability

Data reported in this study may be made available upon request.

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
