# Peer review of "Post Penetrating Keratoplasty Ectasia: Incidence, Risk Factors, Clinical Features, and Treatment Options"

_jcm, 2022, doi:10.3390/jcm11102678_

Round 1
Reviewer 1 Report
Post-Keratoplasty Ectasia YCR Reviewer Comments
This paper is a narrative review of Ectasia developing after Penetrating Keratoplasty, and there are 3 principal explanations presented. I suggest a clearer presentation of evidence for or against the different hypothesis.
The photos have high resolution. these are helpful as models. It would be good if they have a photo of a post-third keratoplasty. This would alert the clinician that the resulting recurrence may be severe enough to indicate another keratoplasty.
The review is comprehensive. But the outcomes of the procedures need to be summarized. For example there are no outcomes for crosslinking in recurrent keratoconus. The authors must state if they did not find published studies on these. This will point out the gaps in the literature.
The number of eyes for the surgical procedures is not in the manuscript. This will inform the clinician of the validity of any conclusion as to efficacy and safety of the procedures mentioned.
I suggest that a summary table be constructed for the Types of surgical procedures, techniques, number of eyes, indications, advantages and disadvantages, and outcomes/results according to the studies cited. These headings are by no means the only information that can populate a summary table.
I also suggest an algorithm for the management of recurrent keratoconus. For example, when should cross-linking be done? What diagnostic tests would indicate changes so that surgery is contemplated? which type of surgery is best for which type of patient. For example, indications may include worsening of the topography on follow-up for 3 months or worsening of visual acuity and developing contact lens intolerance. How much change in topography or visual acuity is enough to warrant a second transplant? These are some of my questions that may or may not be answered by the studies presented.
I suggest the authors include recommendations for unanswered questions like long-term follow-up, careful assessment of the corneal button prior to keratoplasty, and other measures to decrease the risk of a third transplant or any other type of surgical intervention.
Such summary table/s and an algorithm would be helpful for clinicians and would also be an original contribution to the literature.
Author Response
1)This paper is a narrative review of Ectasia developing after Penetrating Keratoplasty, and
there are 3 principal explanations presented. I suggest a clearer presentation of evidence
for or against the different hypothesis.
The mechanism of ectasia post-PK remains unknown. The three principal hypotheses for its development, are reported in the manuscript at chapter 4. In the same chapter we offer a critical analysis of the different hypotheses discussing the limitation of one of them in particular that support the theory that a donor graft affected by keratoconus may be erroneously transplanted in a keratoconus patient.
2)The photos have high resolution. these are helpful as models. It would be good if they
have a photo of a post-third keratoplasty. This would alert the clinician that the resulting
recurrence may be severe enough to indicate another keratoplasty.
Thank you for your suggestion, unfortunately we don’t have an image of a post-third PK ectasia.
3)The review is comprehensive. But the outcomes of the procedures need to be
summarized. For example, there are no outcomes for crosslinking in recurrent keratoconus.
The authors must state if they did not find published studies on these. This will point out
the gaps in the literature.
Thank you for your observation. The outcomes of the procedure has been added to a dedicated table (table 1). In the literature, there are no clinical studies on CXL procedures after PK. We added this information at line 514.
4)The number of eyes for the surgical procedures is not in the manuscript. This will inform
the clinician of the validity of any conclusion as to efficacy and safety of the procedures
mentioned.
Thank for your suggestion. These information have been added to table 1
5)I suggest that a summary table be constructed for the Types of surgical procedures,
techniques, number of eyes, indications, advantages and disadvantages, and
outcomes/results according to the studies cited. These headings are by no means the only
information that can populate a summary table.
Thank you for this suggestion. Summary tables have been added to the manuscript: in table 1 we reported the studies indicating the number of operated eyes and clinical outcomes; in table II we reported advantages and disadvantages of the different techniques reported in this review.
6)I also suggest an algorithm for the management of recurrent keratoconus. For example,
when should cross-linking be done? What diagnostic tests would indicate changes so that
surgery is contemplated? which type of surgery is best for which type of patient. For
example, indications may include worsening of the topography on follow-up for 3 months
or worsening of visual acuity and developing contact lens intolerance. How much change
in topography or visual acuity is enough to warrant a second transplant? These are some
of my questions that may or may not be answered by the studies presented.
Thank for this idea. A possible algorithm for the management of recurrent keratoconus has been added to the manuscript. Regarding cross-linking, we do not feel there is enough evidence so far in the literature to support the use of CXL in post-keratoplasty eyes.
7) I suggest the authors include recommendations for unanswered questions like long-term
follow-up, careful assessment of the corneal button prior to keratoplasty, and other
measures to decrease the risk of a third transplant or any other type of surgical
intervention. Such summary table/s and an algorithm would be helpful for clinicians and would also be an original contribution to the literature.
We agree with this observation. The algorithm that we propose may be a useful tool for the readers for choosing the most appropriate technique of surgical correction. Furthermore the tables that we have added to the manuscript help to summarize the multiplicity of surgical options for the correction of post-PK ectasia. We hope that this may contribute to increase the relevance of this review.
Reviewer 2 Report
Late corneal ectasia following penetrating keratoplasty (PK) for keratoconus can induce increasing irregular astigmatism and result in significant visual impairment, which is a big challenge for corneal surgeons. This review summarizes the epidemiological, clinical and pathogenetic aspects, and currently available options for the correction of the high irregular astigmatism of late recurrence of ectasia after PK. The manuscript is logically clear and well written. There are a few comments as follows:
- The review mainly focuses on corneal ectasia after PK for keratoconus, thus it might be better to change the title to “Post penetrating keratoplasty ectasia” to show the research topic more accurately.
- It lacks Abstract and Key words in the manuscript.
- Since relapse of ectasia after PK has no clear diagnostic criteria, there is still controversy in whether the corneal ectasia after keratoplasty for keratoconus is a recurrence of keratoconus or a primary ectasia of donor cornea. This should be stated in the review.
- It is better to show the epidemiology of corneal ectasia after deep lamellar keratoplasty for keratoconus, and state the differences in the mechanisms of corneal ectasia after these two keratoplasty methods if possible.
- It seems a little bit overwhelming, it is better to condense the content in highly generalized language, and add some presentative pictures of different surgical methods to help readers to understand more easily.
Author Response
1) The review mainly focuses on corneal ectasia after PK for keratoconus, thus it might be better to change the title to “Post penetrating keratoplasty ectasia” to show the research topic more accurately.
Thank you for your suggestion. The title has been modified accordingly: “Post penetrating keratoplasty ectasia: incidence, risk factors, clinical features, and treatment options.”
2) It lacks Abstract and Key words in the manuscript.
Structured abstract and key word have been added to the manuscript
3) Since relapse of ectasia after PK has no clear diagnostic criteria, there is still controversy in whether the corneal ectasia after keratoplasty for keratoconus is a recurrence of keratoconus or a primary ectasia of donor cornea. This should be stated in the review.
This point is extensively discussed in chapter 4 focusing on the hypotheses behind recurrence of ectasia after PK.
4) It is better to show the epidemiology of corneal ectasia after deep lamellar keratoplasty for keratoconus, and state the differences in the mechanisms of corneal ectasia after these two keratoplasty methods if possible.
As reported in the manuscript, ectasia recurrence after DALK is only anecdotally reported. In the manuscript we reported epidemiological data and development hypotheses of post PK ectasia.
5) It seems a little bit overwhelming, it is better to condense the content in highly generalized language, and add some presentative pictures of different surgical methods to help readers to understand more easily.
Correction of post-PK ectasia is a complex theme as well as challenging is its surgical correction. We directly experienced some of the surgical procedures reported in our review and pictures of them are presented in the manuscript. For all the others, we do not have original proprietary operative images and for this reason we referred to the original manuscript for detailed description and images of each procedure. In order to help the readers to navigate through this complex subject we added to the paper 2 tables condensing the results of the surgical procedures with advantages and disadvantages and algorithm for the management of recurrent keratoconus. This should help the readers to better understand the content of this review.
Round 2
Reviewer 2 Report
The authors gave detailed answers to the reviewer's comments. And 2 tables condensing the advantages and disadvantages of different types of surgical procedures for the correction of post-penetrating keratoplasty ectasia have been added to the manuscript, which could help the readers to better understand the content of this review.